# Oral Conditions and Oral Health-Related Quality of Life of People with Ehlers-Danlos Syndromes (EDS): A Questionnaire-Based Cross-Sectional Study

**DOI:** 10.3390/medicina56090448

**Published:** 2020-09-04

**Authors:** Marcel Hanisch, Moritz Blanck-Lubarsch, Lauren Bohner, Dominik Suwelack, Johannes Kleinheinz, Jeanette Köppe

**Affiliations:** 1Department of Cranio-Maxillofacial Surgery, Research Unit Rare Diseases with Orofacial Manifestations, University Hospital Münster, Albert-Schweitzer-Campus 1, Gebäude W 30, D-48149 Münster, Germany; lauren.bohner@ukmuenster.de (L.B.); johannes.kleinheinz@ukmuenster.de (J.K.); 2Department of Oral Surgery and Dental Emergency Care, Faculty of Health, Witten/Herdecke University, Alfred-Herrhausen-Strasse 45, 58455 Witten, Germany; 3Department of Orthodontics, University Hospital Münster, Albert-Schweitzer-Campus 1, Gebäude W 30, D-48149 Münster, Germany; moritz.blanck-lubarsch@ukmuenster.de; 4Department of Prosthetic Dentistry and Biomaterials, University Hospital Münster, Albert-Schweitzer-Campus 1, Gebäude W 30, D-48149 Münster, Germany; dominik.suwelack@ukmuenster.de; 5Institute of Biostatistics and Clinical Research, University of Münster, Schmeddingstrasse 56, D-48149 Münster, Germany; jeanette.koeppe@ukmuenster.de

**Keywords:** rare diseases, oral health related quality of life, oral manifestations, Ehlers–Danlos syndromes, patient-related outcome

## Abstract

*Background and objective*: To date, there have only been a few studies on oral health-related quality of life (OHRQoL) of people with Ehlers–Danlos syndromes (EDS) and oral conditions. The aim of this study was, therefore, to analyze the OHRQoL of people with EDS from their own point of view as well as obtain information about their age at the time of the diagnosis, the period of time until diagnosis, and the presence of oral conditions (if any) and their association with oral health quality. *Methods:* The study was designed as an anonymous questionnaire-based cross-sectional study. We conducted a descriptive analysis of the Oral Health Impact Profile-14 (OHIP-14) scores, age of the participants, age at diagnosis, and the time-period between the first signs of the disease and the diagnosis of EDS. To verify the differences in OHIP-14 scores between patients with and without oral conditions, a Mann–Whitney U test was performed. A multivariate quantile (median) regression analysis was performed to evaluate the effect of different general characteristics (gender, age, and the presence of oral conditions) on the OHIP 14 scores. Furthermore, using a Mann–Whitney U test, the influence of different oral conditions was verified by testing the differences between patients without any oral conditions and patients with a specific diagnosis. *Results:* A total of 79 evaluable questionnaires from 66 female (83.5%) and 13 male (16.5%) participants were analyzed. On average, after the first condition, it takes 18.36 years before EDS are correctly diagnosed. Oral conditions were described by 69.6% of the participants. The median (interquartile range) OHIP-14 score was eight (ten) points for patients without oral conditions and 19 (15) for patients with oral conditions. The multivariable quantile regression shows a statistical notable association between OHIP-14 score and oral conditions (*p* < 0.001). OHIP-14 scores for dysgnathia, periodontitis, TMD (Temporomandibular dysfunction), a high-arched palate, malocclusion, and the anomaly of tooth formation were statistical notably different between the participants with and the participants without oral conditions. *Conclusions:* Long diagnostic pathways seem to be a typical problem in patients with EDS. Oral conditions associated with the underlying disease occurred regularly and showed a negative correlation with OHRQoL.

## 1. Introduction

Ehlers–Danlos syndromes (EDS) are a clinically and genetically heterogeneous group of hereditary connective tissue disorders involving joint hyperlaxity, cutaneous hyperelasticity, and tissue fragility [1,2]. A revised classification containing 13 subtypes was published by the International EDS Consortium, with a number of clinical conditions to guide and improve the diagnosis of each subtype [2]. In the European Union, a disease is considered ‘rare’ if it affects fewer than one in two thousand people [3]. In the case of EDS, the prevalence varies, depending on the type: between 1:30,000 (classical type) and < 1:1,000,000 (arthrochalasia type), whereas for the rarest form (the periodontitis type), the prevalence is unknown [1].

EDS’ oral conditions have been described as periodontitis [2,4,5], temporomandibular dysfunctions [2,6], bleeding tendencies [7,8], enamel hypoplasia [2,9], shape abnormalities of the teeth and changes in the number of teeth [2,8], a high palate [2,10], dysgnathia, and malocclusion [2,11], as well as decreased effects of local anesthesia [12,13]. 

It is known that periodontal diseases and temporomandibular joint dysfunctions have a negative impact on quality of life [14,15,16].

To date, there have been only a few studies on the oral health-related quality of life (OHRQoL) of people with EDS who report reduced OHRQoL [6,17,18]. A proven method for evaluating OHRQoL is the Oral Health Impact Profile-14 (OHIP-14) questionnaire (Appendix A), which measures the frequency of 14 different functional and psychosocial factors influencing OHRQoL [19].

The maximum number of 56 point in the OHIP-14 version corresponds to a very high impact of patient’s OHRQoL [12]. Therefore, the OHIP-14 score is a suitable instrument for describing the influence of oral health on general health from a patient’s perspective. 

Reissmann et al. [20] have investigated how many events are necessary for one more OHIP points to be added by participants. The OHIP-49 questionnaire was used for their study, which does not allow for a direct comparison to the present study. However, and nevertheless, it does give an impression of the considerably reduced OHRQoL. They concluded that, on average, a total of 15.2 events are required in a month for the OHIP total score to increase by one point, i.e., a person must interrupt a meal, feel unwell or have pain on 15.2 times per month for the OHIP score to rise by one point. This means that the person must struggle with these constraints every two days.

The aim of the study was to analyze the OHRQoL of people with EDS and the association between the presence of oral conditions (if any) and the reported oral health quality. Therefore, we hypothesized that participant’s reporting any oral conditions have a higher OHIP-14 score compared to those who did not report any conditions. Furthermore, the influence of different conditions on the oral health quality, obtained information of age, time of diagnosis and the period until diagnosis were analyzed.

## 2. Methods

### 2.1. Study Design

The study was designed as an anonymous questionnaire-based cross-sectional study involving people with EDS to evaluate their respective OHRQoL.

A questionnaire consisting of free text questions about age, gender, oral conditions (if any), and the period between the first symptom of the disease and its diagnosis was developed for this purpose. In addition, the standardized German version of the OHIP-14 questionnaire was added to the evaluation of the OHRQoL [21].

Each of the 14 questions in the OHIP-14 questionnaire was assigned standardized numerical values: 0 = never, 1 = hardly ever, 2 = sometimes, 3 = often and 4 = very frequently. It was possible to score a total of zero and a maximum of 56 points. The higher the score, the worse the OHRQoL. The questions referred to experiences registered over the past month. 

The study was presented at the annual meeting of the ‘Ehlers–Danlos initiative e.V.’ self-help group on 22 September 2018, in Bad Kissingen, Germany. The participants were asked to send the completed questionnaire, which was provided online to the members by the self-help group either by email or by post to the study organizers. Responses were taken into account until 22 March 2019.

This study was approved by the Ethics Committee of the Medical Association of Westphalia Lippe and the Westphalian Wilhelms—University of Münster (Ref. No. 2016 006 f S).

### 2.2. Participants

People aged 16 years and above and are affected by EDS in the Federal Republic of Germany were eligible to participate. The number of eligible persons affected was specified by the ‘Deutsche Ehlers–Danlos Initiative e.V.’, which consisted of 240 members.

### 2.3. Data Source

In addition to age, sex and disease, possible oral conditions, age at the time of diagnosis and the period from the first appearance of the disease until diagnosis was recorded, the individual OHIP values were calculated. 

### 2.4. Statistical Analysis

SAS 9.4 (SAS Institute, Cary, NC, USA) and R version 3.1.6 (R Foundation for Statistical Computing, Vienna, Austria) were used for statistical analysis. All the analyses were exploratory, not confirmatory, and adjustments for multiple testing were not performed. 

A descriptive analysis of the OHIP-14 scores, participants’ ages, age at diagnosis, and time-period between the first signs of the disease and the diagnosis of EDS was conducted by determining the mean, median, minimum, maximum, standard deviation (STD), and interquartile range (IQR) for both the males and the females. To verify if there were differences between the OHIP-14 scores of patients with and patients without oral conditions, a Mann–Whitney U test was performed. A multivariate quantile (median) regression analysis was performed to evaluate the effect of different general characteristics (gender, age, and the presence of oral conditions) on the OHIP-14 scores. Furthermore, using a Mann–Whitney U test, the influence of different oral conditions was verified by testing if there were differences between patients without any oral conditions and patients with a specific diagnosis.

The researchers compared participants who reported having EDS-related conditions to participants who reported having none; this practice was in favor of comparing two subgroups that declared being symptomatic or asymptomatic. The evaluation was completed this way, as the respondents may have had different combinations of symptoms/conditions and the control group would have been too heterogeneous (i.e., including participants having no conditions and participants having other oral conditions). However, comparisons between individual conditions were not useful for the present cohort.

## 3. Results

### 3.1. Participants

A total of 79 evaluated questionnaires from 66 female (83.5%) and 13 male (16.5%) participants were analyzed. The gender-independent median (interquartile range (IQR)) of age was 38 (25) years (range: 16–81 years). The median (IQR) age for the female participants was 38 (26) years and the median (IQR) age of the male participants was 42 (29) years.

### 3.2. Age at Diagnosis and Time Until Diagnosis

The median (IQR) age at which EDS was diagnosed was 29 (±24) years for females and 28 (±36) years for males. The mean (standard deviation (STD)) period of time from the first signs of the disease to diagnosis was 19.35 (±18) years (range 0–77 years) for the females, 13.15 (±16) years (range 0–42 years) for the males.

### 3.3. Reported Oral Conditions

When asked about oral conditions, 79 participants provided responses, 55 (69.6%) of which described oral conditions. Thirty-seven respondents cited oral conditions as various forms of skeletal malocclusion and growth aberrations (e.g., high palate, micrognathia, macrognathia, microgenia or ‘jaws do not fit together’), while a high palate was explicitly mentioned by 18 respondents. Tooth-related dysgnathia (malocclusions) were also described by 16 participants. There were also combined forms of skeletal and dentoalveolar dysgnathia among the respondents. Craniomandibular dysfunctions occurred in 21 of the respondents, according to the data. Gingivitis/periodontitis was also mentioned as oral conditions by 17 respondents, shape abnormalities of the teeth were described by 11 respondents, and a decrease in the number of teeth (hypodontia/oligodontia) was described by a total of 6 respondents. Figure 1 provides an overview of the conditions described by the respondents.

### 3.4. OHIP-14 Scores

In the group of participants without any oral conditions, a median (IQR) score of 8 (10) points was found. In contrast, in the group of participants with reported oral conditions, the median of the OHIP-14 score was 19 (15) points (*p* < 0.001).

Furthermore, the influence of gender, age and the presence of oral conditions were analyzed by multivariate quantile regression. However, age and gender did not seem to influence the OHIP 14 score. The resulting coefficients can be found in Table 1.

### 3.5. Influence of OHIP-14 Score on the Attendance of Oral Conditions

All the oral conditions that were named by more than 10 respondents were further analyzed. The OHIP-14 scores for these conditions and for the group of participants without oral conditions are plotted in Figure 2.

It is remarkable that for all the conditions that were considered, the median score was statistically notably higher than the score for participants without oral conditions (all the *p*-values were sufficiently small). In Table 2, the median, mean, and *p*-values of the described comparisons are given for all the conditions considered.

## 4. Discussion

### Strength and Limitations of the Study

The oral conditions described by the respondents in this study are manifestations that are already known, which may occur in EDS. However, the correct terminology cannot always be expected because the conditions have not been clinically verified. When categorizing the conditions, the information provided by the participants translated and assigned in a medical term accordingly. When categorizing the conditions, the information provided by the participants was translated and assigned to a corresponding medical term.

For example, some study participants referred to the condition ‘upper jaw too small’, which was recently assigned to the term ‘micrognathia’ in the ‘dysgnathia’ category. Other participants only stated the condition ‘the jaws don’t fit together’, which was also assigned to the ‘dysgnathia’ category. Furthermore, the sample population is too heterogeneous and each symptom may have a different influence on the OHRQoL throughout the life. For instance, a “micrognathia” may involve esthetic concerns in a person of 18 years, but not in a patient of 81 years. On the contrary, a craniomandibular dysfunction may be not noticed in young age, whereas it could be relevant for senior patients. However, the negative influence of temporomandibular dysfunctions on the quality of life is generally known [14]. Moreover, the perception of “micrognathia” is influenced from self-esteem, and it does not necessarily match with the severity of micrognathia [22]. The detailed assignment of oral condition must, therefore, be regarded as having certain limitations. Therefore, oral conditions should be evaluated clinically in future studies.

Moreover, the analysis does not account for confounding factors; this may be a significant bias for some symptoms, such as “gingivitis, periodontitis [23].

Moreover, we don’t know if the self-consciousness of syndromic status may enhance the perception of any oral symptom as “abnormalities” or “invalidating factor”. For instance, we don’t know the clinical parameters (periodontal pocket deep (PPD), bleeding on probing (BoP), gingival index (GI), periodontitis index (PI)) of the patients with “periodontitis, gingivitis”; therefore, we don’t know if a person who is not affected with EDS, with the same clinical parameters, would have the same OHIP-14 score as an EDS patient [24,25]. However, the negative influence of periodontal diseases on the quality of life is generally known [15].

However, it was possible to evaluate the data of 79 participants, which is a relatively high sample size.

The literature has described a fundamental problem concerning the period of time until the diagnosis of rare diseases, in this case, seven years on average [26]. In our group of respondents, a period of 18.36 years between the onset of the first symptom and correct diagnosis of the disease was described by those affected with EDS, regardless of gender. Those affected were on average 28.5 years old. We could identify only one study covering the age of diagnosis of people affected by EDS [27]. The study found that 28 years passed between the first symptom and the diagnosis of EDS, after consulting 16 different doctors. Our results thus confirm the problem of significant delays in diagnosing EDS. It, therefore, seems obvious that those affected by EDS will have to endure a period of unnecessary suffering before finally being diagnosed. 

Although women seem to be slightly more likely to be affected by EDS than men, it does not explain the ratio of 83.5% to 16.5% in our study. We explain this ratio with two factors: women are more likely to participate in self-help groups than men, and are more willing to participate in studies than men. We have already noticed this in other studies on rare diseases [28,29,30].

The partial overlap of different phenotype features between the 13 subtypes of EDS contributes to the difficulty of diagnosis [31]. The symptoms are also sometimes misinterpreted as ‘psychosomatic’ or ‘nonexistent’ [32].

To speed up the diagnostic process in the future, medical education, publications, and guidelines must inform medical doctors and dentists about the disease and its symptoms/conditions.

Dentists who correctly identify oral conditions can significantly contribute to earlier diagnosis of the underlying disease.

In general, our study showed that OHIP values among participants who described oral conditions and those who did not report oral conditions were statistically noticeable. For example, the OHIP values for those with oral conditions were, on average, 21.35 points. While for those with no oral conditions, the average was 10.13 points. 

On average, the OHIP-14 overall score of all the participants was 18.25 points. This shows on the one hand that oral conditions contribute to a reduced OHRQoL, which is of course understandable. However, it also shows that people with EDS, even without oral conditions, show a worse OHRQoL compared to the German average population (4.09 points) [33]. When oral conditions are reported, the affected people show a significantly reduced OHRQoL, and the suffering can be understood by the interpretation of an OHIP point as described in the introduction section [20]. Thus, it can be stated that those affected with EDS are exposed to considerable suffering pressure. Consequently, they should receive support from the health care system (e.g., physiotherapy) without bureaucratic obstacles.

A Swedish study [17] reported an average overall score of 11.1 points regardless of gender and a total score of 11.8 points among female participants who made up the majority of the study. Thus, the values in the present study were above the results obtained by Berglund [17], who reported worse OHRQoL among people with EDS compared to the standard population.

In a study by John et al. [31], an average OHIP-14 score of 4.09 points was determined for the German standard population. Thus, these results show that, as also determined by Berglund et al. in Sweden [17], people with EDS have a worse OHRQoL than the general population. Other studies have also shown a worse OHRQoL for rare diseases [34]. The results obtained suggest that people with EDS, especially when they have oral manifestations, require additional support from the health-care system to improve their oral health and the resulting quality of life. In practice, this might mean, for example, additional services provided free-of-charge by the health-care system. The data obtained here, thus, reflects a possible problem of health-care provision. As the participants were not examined clinically, this should be considered a limitation of this study. The aim of future studies should be to clinically objectify the subjective assessment of the OHRQoL of those affected.

## 5. Conclusions

Long diagnostic pathways seem to be a typical problem in patients with EDS. The OHIP-14 scores are higher, and therefore, are associated with a reduced OHRQoL than known of in the normal population. EDS is serious, it has serious oral consequences, and it has a negative association with OHRQoL, as shown in this study. Thus, it can be stated that those affected with EDS are exposed to considerable suffering pressure. Consequently, they should receive additional support (e.g., physiotherapy) from their national health care system.

## Figures and Tables

**Figure 1 medicina-56-00448-f001:**
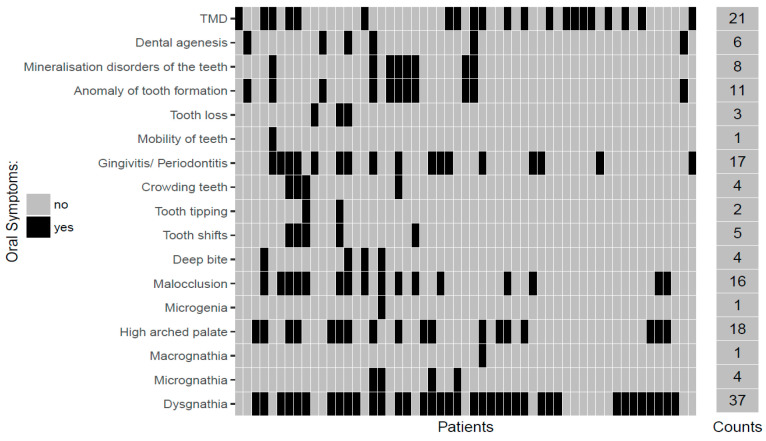
Oral conditions described by the participants.

**Figure 2 medicina-56-00448-f002:**
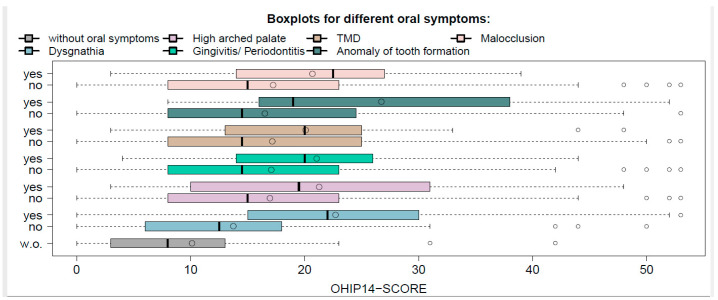
Boxplots of OHIP-14 score for different oral conditions.

**Table 1 medicina-56-00448-t001:** Quantile regression of Oral Health Impact Profile-14 (OHIP-14) score depending on different general characteristics.

	Regression Coefficient	95%-Confidence Interval	*p*-Values
Age	0.13	(−0.07; 0.33)	0.191
Female Sex	−0.53	(−7.45; 6.38)	0.878
Oral conditions	9.60	(4.18; 15.02)	<0.001 *

* Means were statistically noticeable.

**Table 2 medicina-56-00448-t002:** Median and mean of the OHIP-14 score for different oral conditions. *p*-values are given for testing the difference between participants with the corresponding condition and participants without any oral conditions.

Conditions	Number of Patients	Median (IQR)		Mean (STD)		*p*-Value
		Yes	No	Yes	No	
Dysgnathia	37	22 (15)	12.5 (12)	22.7 (12.3)	13.7 (11.7)	<0.001
Periodontitis	17	20 (12)	14.5 (15)	21.1 (10.0)	17.1 (13.3)	<0.001
TMD	21	20 (12)	14.5 (17)	20.1 (12.2)	17.2 (13.0)	0.002
High arched palate	18	19.5 (21)	15 (15)	21.3 (12.2)	17.0 (12.8)	0.002
Malocclusion	16	22.5 (13)	15 (15)	20.1 (9.3)	17.2 (13.5)	0.001
Anomaly of tooth formation	11	19 (28)	14.5 (16.5)	26.7 (15.5)	16.5 (11.8)	<0.001

IQR: interquartile range; STD: standard deviation; TMD: Temporomandibular dysfunction.

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
