# Peer review of "Oral Conditions and Oral Health-Related Quality of Life of People with Ehlers-Danlos Syndromes (EDS): A Questionnaire-Based Cross-Sectional Study"

_medicina, 2020, doi:10.3390/medicina56090448_

Round 1
Reviewer 1 Report
The paper could use significant reorganization/re-framing. The main point seems to be that those with ED and oral symptoms have impacts upon OHRQoL. Readers may ask "so what?" because that point seems obvious. I believe a stronger message may be the extent to which patients with ED have oral symptoms and the extent to which these impact their OHRQoL- which in turn is a better argument for why dentists should be aware of these syndromes and be helpful in obtaining earlier diagnoses.
Oral symptoms of ED as described are really more oral conditions because they are anatomical or structural changes as a result of the syndrome. The word symptoms implies a more transient or developing situation such as pain, dry mouth, gingivitis, etc.
For analysis, it may be informative to look at OHRQoL stratified by ED conditions that effect the jaw vs. the teeth. You mention this in the results but don't take it any further.
Introduction & Methods: A more complete description of OHRQoL measurement is needed. Perhaps you may even want list out each question in the OHIP-14 (int he section labeled data source). How many surveys were sent out? For self report of oral conditions-how was this reported? Did participants proved open-ended responses or check from a list of conditions? IF checking from a list, did you provide them with the scientific term or a more common name for conditions? The introduction should state a bit more about the importance of OHRQoL.
Results: What was your response rate? Is there reason to believe that those that responded were systematically different than those that did not respond? Was your distribution of males and females similar to the prevalence in the general population of people diagnosed with ED? In figure one you may want to describe the terms for dental conditions for a non-dental audience. Perhaps order the conditions from least to most prevalent in order to make the figure easier to interpret? For table one, noted with the asterisk, do you means significance rather than conspicuous? Figure three is difficult to interpret, perhaps just show the "yes" responses?
Discussion: More clearly state strengths and weaknesses. As stated earlier, your main point should me more clearly framed and then carried through to the discussion. Your discussion focuses mainly on time to diagnosis, but that seems secondary and not very related to your methods and results. Do not restate results in your discussion (see lines 197-200). Make sure to clearly state what makes your research unique or new.
Conclusion should be much stronger: ED is serious, it has serious oral consequences and OHRQoL as shown by your research....
Figure 2 seems unnecessary-perhaps replace with frequency of the 14 OHRQoL impacts?
Discussion: Clearly denote self report of oral conditions as a limitation.
Author Response
We would like to thank the Editor and the Reviewers for revising our manuscript [medicina-885699] entitled “Oral symptoms and oral health-related quality of life of people with Ehlers-Danlos syndromes (EDS): A questionnaire-based cross-sectional study“ and the constructive points discussed. The helpful comments and suggestions for improving the manuscript have been incorporated into the revised version and all changes were highlighted “Track Changes”. In this letter, we provide a point-by-point response to each addressed comment and hope the manuscript is now suitable for publication in Medicina.
Reviewer 1
Comments and Suggestions for Authors
The paper could use significant reorganization/re-framing. The main point seems to be that those with ED and oral symptoms have impacts upon OHRQoL. Readers may ask "so what?" because that point seems obvious. I believe a stronger message may be the extent to which patients with ED have oral symptoms and the extent to which these impact their OHRQoL- which in turn is a better argument for why dentists should be aware of these syndromes and be helpful in obtaining earlier diagnoses.
Answer: We compared participants who reported having EDS-related conditions to participants who reported having none; this practice was in favour of comparing two subgroups that declared being symptomatic or asymptomatic. The evaluation was completed this way, as the respondents may have had different combinations of symptoms/conditions and the control group would have been too heterogeneous (i.e., including participants having no conditions and participants having other oral conditions). However, comparisons between individual conditions were not useful for the present cohort (line 123-129).
Including all conditions into one regression model, to adjust the influence and evaluate possible interactions (i.e. worse combinations of different conditions) would give a closer impression how the OHRQoL is influenced by the different disease. However, the sample size of this study is too small for such an analysis. Therefore, we could only state about the univariate influence of the conditions, which is presented in figure 3.However, we will take this information into account for future studies. Here, it would be possible to conduct another targeted clinical investigation to determine which EDS subtypes occur with which oral conditions. In order to show a correlation between oral conditions and OHRQoL, we have not included enough participants of the individual subtypes here.
Oral symptoms of ED as described are really more oral conditions because they are anatomical or structural changes as a result of the syndrome. The word symptoms implies a more transient or developing situation such as pain, dry mouth, gingivitis, etc.
Answer: we have now replaced "symptoms" with "conditions" in the manuscript
For analysis, it may be informative to look at OHRQoL stratified by ED conditions that effect the jaw vs. the teeth. You mention this in the results but don't take it any further.
Answer: We compared participants who reported having EDS-related conditions to participants who reported having none; this practice was in favour of comparing two subgroups that declared being symptomatic or asymptomatic. The evaluation was completed this way, as the respondents may have had different combinations of symptoms/conditions and the control group would have been too heterogeneous (i.e., including participants having no conditions and participants having other oral conditions). However, comparisons between individual conditions were not useful for the present cohort (line 123-129).
Thank you for this comment. You are right, it could be also interesting to analyze the subgroups given by the conditions effected the jaw and teeth separately. However, we decided to analyze the conditions, which are reported by at least 10 participants (see figure 3) to get a closer look on the influence of the most reported conditions also with regard to the fact that the symptoms are not clinically verified.
Introduction & Methods: A more complete description of OHRQoL measurement is needed. Perhaps you may even want list out each question in the OHIP-14 (int he section labeled data source).
Answer: we have now attached the ohip-14 questionnaire as a supplement.
How many surveys were sent out?
Answer: we have made the questionnaire available online on the website of the self-help group. Therefore, we cannot give any information on the number of people, who had access to the questionnaire. However, the self-help group had 240 members at that time.
.
For self report of oral conditions-how was this reported? Did participants proved open-ended responses or check from a list of conditions? IF checking from a list, did you provide them with the scientific term or a more common name for conditions?
Answer: we asked the participants to make statements in their own words about their conditions or symptoms in the tooth, mouth and jaw area. These statements were translated by us into medical terms. We are aware that the symptoms were not clinically evaluated. We also formulated this in the discussion as a limitation of the study and made the recommendation to evaluate this oral conditions clinically in future studies. (line 202-212)
The introduction should state a bit more about the importance of OHRQoL.
Answer: we have now added more information about OHRQoL: “The maximum number of 56 point in the OHIP-14 version corresponds to a very high impact of patient’s OHRQoL [12]. Therefore, the OHIP-14 score is a suitable instrument for describing the influence of oral health on general health from a patient’s perspective.
Reissmann et al. [17] have investigated how many events are necessary for one more OHIP points to be added by participants. The OHIP-49 questionnaire was used for their study, which does not allow for a direct comparison to the present study.But, nevertheless, it does give an impression of the considerably reduced OHRQoL. They concluded that, on average, a total of 15.2 events are required in a month for the OHIP total score to increase by one point. I.e., a person must interrupt a meal, feel unwell or have pain on 15.2 times per month for the OHIP score to rise by one point. This means that the person must struggle with this constraints every two days.” (line 68-77)
Results: What was your response rate? Is there reason to believe that those that responded were systematically different than those that did not respond? Was your distribution of males and females similar to the prevalence in the general population of people diagnosed with ED?
- Answer: we have made the questionnaire available online on the website of the self-help group. Therefore we cannot give any information on how many people were reached with it or what the response rate was. The self-help group had 240 members at that time. Although women seem to be slightly more likely to be affected by EDS than men, it obviously does not explain the ratio of 83.5% to 16.5% in our study. We explain this ratio with two factors: women are more likely to participate in self-help groups than men and are more willing to participate in studies than men. We have already noticed this in other studies on rare diseases [ Wiemann, S.; Frenzel Baudisch, N.; Jordan, R.; Kleinheinz, J.; Hanisch, M. Oral Symptoms and Oral Health-Related Quality of Life in People with Rare Diseases in Germany: A Cross-Sectional Study. J. Environ. Res. Public. Health 2018, 15 (7), 1493. https://doi.org/10.3390/ijerph15071493.
- Hanisch, M.; Wiemann, S.; Bohner, L.; Kleinheinz, J.; Jung, S. Association between Oral Health-Related Quality of Life in People with Rare Diseases and Their Satisfaction with Dental Care in the Health System of the Federal Republic of Germany. J. Environ. Res. Public. Health 2018, 15 (8), 1732. https://doi.org/10.3390/ijerph15081732.
- Hanisch, M.; Sielker, S.; Jung, S.; Kleinheinz, J.; Bohner, L. Self-Assessment of Oral Health-Related Quality of Life in People with Ectodermal Dysplasia in Germany. J. Environ. Res. Public. Health 2019, 16 (11), 1933. https://doi.org/10.3390/ijerph16111933.].
- We have now added this in the discussion section (line 230-233).
In figure one you may want to describe the terms for dental conditions for a non-dental audience. Perhaps order the conditions from least to most prevalent in order to make the figure easier to interpret? For table one, noted with the asterisk, do you means significance rather than conspicuous? Figure three is difficult to interpret, perhaps just show the "yes" responses?
Answer: In figure one, we sorted the conditions by the number of reports. In table one, we changed “conspicuous” to “noticeable”. From a statistical point of few, the term “significant” is only meaningful in the context of a confirmatory study and should, strictly speaking, only be used there. Thus, we used “statistical noticeable” instead, since our study was purely explorative.
Discussion: More clearly state strengths and weaknesses. As stated earlier, your main point should me more clearly framed and then carried through to the discussion. Your discussion focuses mainly on time to diagnosis, but that seems secondary and not very related to your methods and results. Do not restate results in your discussion (see lines 197-200). Make sure to clearly state what makes your research unique or new.
Answer: we asked the participants to make statements in their own words about their conditions or symptoms in the tooth, mouth and jaw area. These statements were translated by us into medical terms. We are aware that the symptoms were not clinically evaluated. We also formulated this in the discussion as a limitation of the study and made the recommendation to evaluate this oral conditions clinically in future studies. (Line 193-212)
We have now deleted line 197-200.
We rephrased the discussion section and added this paragraph: “I In general, our study showed that OHIP values among participants who described oral conditions and those who did not report oral conditions were statistically noticeable. For example, the OHIP values for those with oral conditions were, on average, 21.35 points. While for those with no oral conditions, the average was 10.13 points.
On average, the OHIP-14 overall score of all the participants was 18.25 points. This shows on the one hand that oral conditions contribute to a reduced OHRQoL, which is of course understandable. However, it also shows that people with EDS, even without oral conditions, show a worse OHRQoL compared to the German average population (4.09 points). [30] When oral conditions are reported, the affected people show a significantly reduced OHRQoL, and the suffering can be understood by the interpretation of an OHIP point as described in the introduction section. [17] Thus, it can be stated that those affected with EDS are exposed to considerable suffering pressure. Consequently, they should receive support from the health care system (e.g. physiotherapy) without bureaucratic obstacles.” (line 242-254)
Conclusion should be much stronger: ED is serious, it has serious oral consequences and OHRQoL as shown by your research....
Answer: We rephrased the conclusion section: “…Long diagnostic pathways seem to be a typical problem in patients with EDS. EDS is serious, it has serious oral consequences and has a negative association with OHRQoL, as shown in this study. Thus, it can be stated that those affected with EDS are exposed to considerable suffering pressure. As a consequence, they should receive additional support from their national health care system.…” (line 271-274)
Figure 2 seems unnecessary-perhaps replace with frequency of the 14 OHRQoL impacts?
Answer: Thank you for the comment. To shorten up the results, we removed figure 2 from the manuscript.
Discussion: Clearly denote self report of oral conditions as a limitation.
Answer: We formulated this in the discussion as a limitation of the study and made the recommendation to evaluate this oral conditions clinically in future studies. (line 193-212)
Reviewer 2 Report
Dear authors,
the study is worthy of publication. I have some minor comments to improve the paper.
RESULTS:
page 3, lines 124.128: I would rephrase this sentences, and remove the abbreviations:"The median age at which EDS was diagnosed was 29 (±24) years for females and 28 (±36) years for males. The mean (STD) period of time from the first signs of the disease to diagnosis was 19.35 (±18) years (range 0–77 years) the females, 13.15 (±16) years (range 0–42 years) for males."I think that gender-independent value should be removed, as it does not add any relevant information. I have also rounded up the values of the SD.
page 3, lines 130-133:
DISCUSSION:
you should add a paragraph with strenghts and limitations of the study.
As a limitation, you can say that sample population is too heterogeneous that each symptom may have a different influence on the Qol throughout the life. For instance, a "micrognathia" may involve esthetic concerns in a person of 18 years, but not in a patient of 81 years. On the contrary, a TMD symptom may be not noticed in young age, whereas it could be relevant for senior patients.
Moreover, the perception of "micrognathia" is influenced from self-esteem, and it does not necessarily match with the severity of micrognathia. As a suggestion, please add the reference below: Staderini E, De Luca M, Candida E, et al. Lay People Esthetic Evaluation of Primary Surgical Repair on Three-Dimensional Images of Cleft Lip and Palate Patients. Medicina (Kaunas). 2019;55(9):576. Published 2019 Sep 8. doi:10.3390/medicina55090576
Moreover, the analysis does not account for confounding factors; this may be a significant bias for some symptoms, such as "gingivitis, periodontitis". As a suggestion, please add the reference below: Patini R, Staderini E, Camodeca A, Guglielmi F, Gallenzi P. Case Reports in Pediatric Dentistry Journals: A Systematic Review about Their Effect on Impact Factor and Future Investigations. Dent J (Basel). 2019;7(4):103. Published 2019 Oct 24. doi:10.3390/dj7040103
Moreover, we don't know if the self-consciousness of syndromic statu may enhance the perception of any oral symptom as "abnormalities" or "invalidating factor". For instance, we don't know the clinical parameters (PPD, BoP, GI, PI) of the patients with "periodontitis,gingivitis"; therefore, we don't know if a normal person, with the same clinical parameters, would have the same OHIP-14 score as an EDS patient. As a suggestion, please add the references below:
Guglielmi F, Staderini E, Iavarone F, Di Tonno L, Gallenzi P. Zimmermann-Laband-1 Syndrome: Clinical, Histological, and Proteomic Findings of a 3-Year-Old Patient with Hereditary Gingival Fibromatosis. Biomedicines. 2019;7(3):48. Published 2019 Jun 29. doi:10.3390/biomedicines7030048
Staderini E, Guglielmi F, Cordaro M, Gallenzi P. Ossifying epulis in pseudohypo-parathyroidism: a case-based therapeutic approach. Eur J Paediatr Dent. 2018;19(3):218-220. doi:10.23804/ejpd.2018.19.03.9
REFERENCES:
references should be formatted according to the journal guidelines.
Author Response
Dear authors,
the study is worthy of publication. I have some minor comments to improve the paper.
RESULTS:
page 3, lines 124.128: I would rephrase this sentences, and remove the abbreviations:"The median age at which EDS was diagnosed was 29 (±24) years for females and 28 (±36) years for males. The mean (STD) period of time from the first signs of the disease to diagnosis was 19.35 (±18) years (range 0–77 years) the females, 13.15 (±16) years (range 0–42 years) for males." I think that gender-independent value should be removed, as it does not add any relevant information. I have also rounded up the values of the SD. page 3, lines 130-133:
Answer: We have rephrased the sentence according to your recommendations. (line 137-140)
DISCUSSION:
you should add a paragraph with strenghts and limitations of the study.
As a limitation, you can say that sample population is too heterogeneous that each symptom may have a different influence on the Qol throughout the life. For instance, a "micrognathia" may involve esthetic concerns in a person of 18 years, but not in a patient of 81 years. On the contrary, a TMD symptom may be not noticed in young age, whereas it could be relevant for senior patients.
Moreover, the perception of "micrognathia" is influenced from self-esteem, and it does not necessarily match with the severity of micrognathia. As a suggestion, please add the reference below: Staderini E, De Luca M, Candida E, et al. Lay People Esthetic Evaluation of Primary Surgical Repair on Three-Dimensional Images of Cleft Lip and Palate Patients. Medicina (Kaunas). 2019;55(9):576. Published 2019 Sep 8. doi:10.3390/medicina55090576
Answer: we included a paragraph with the limitations of our study and included the recommended literature:
“The oral conditions described by the respondents in this study are manifestations that are already known, which may occur in EDS. However, the correct terminology cannot always be expected because the conditions have not been clinically verified. When categorising the conditions, the information provided by the participants translated and assigned in a medical term accordingly. When categorising the conditions, the information provided by the participants was translated and assigned to a corresponding medical term. For example, some study participants referred to the condition ‘upper jaw too small’, which was recently assigned to the term ‘micrognathia’ in the ‘dysgnathia’ category. Other participants only stated the condition ‘the jaws don’t fit together’, which was also assigned to the ‘dysgnathia’ category. Furthermore, the sample population is too heterogeneous and each symptom may have a different influence on the OHRQoL throughout the life. For instance, a "micrognathia" may involve esthetic concerns in a person of 18 years, but not in a patient of 81 years. On the contrary, a craniomandibular dysfunction may be not noticed in young age, whereas it could be relevant for senior patients. Moreover, the perception of "micrognathia" is influenced from self-esteem, and it does not necessarily match with the severity of micrognathia. [19] The detailed assignment of oral condition must, therefore, be regarded as having certain limitations. As a consequence, oral conditions should be evaluated clinically in future studies.” (line 194-212)
Moreover, the analysis does not account for confounding factors; this may be a significant bias for some symptoms, such as "gingivitis, periodontitis". As a suggestion, please add the reference below: Patini R, Staderini E, Camodeca A, Guglielmi F, Gallenzi P. Case Reports in Pediatric Dentistry Journals: A Systematic Review about Their Effect on Impact Factor and Future Investigations. Dent J (Basel). 2019;7(4):103. Published 2019 Oct 24. doi:10.3390/dj7040103
Answer: We evaluate possible confounding factors by analyzing the univariate association of different oral conditions and the OHIP-14 score, see figure 2. Including all conditions into one regression model, to adjust the influence and evaluate possible interactions (i.e. worse combinations of different conditions) would give a closer impression how the OHRQoL is influenced by the different disease. However, the sample size of this study is too small for such an analysis. Therefore, we could only state about the univariate influence of the conditions, which is presented in figure 2 (see Reviewer 1).
Moreover, we don't know, if the self-consciousness of syndromic status may enhance the perception of any oral symptom as "abnormalities" or "invalidating factor". For instance, we don't know the clinical parameters (PPD, BoP, GI, PI) of the patients with "periodontitis,gingivitis"; therefore, we don't know if a normal person, with the same clinical parameters, would have the same OHIP-14 score as an EDS patient. As a suggestion, please add the references below:
Guglielmi F, Staderini E, Iavarone F, Di Tonno L, Gallenzi P. Zimmermann-Laband-1 Syndrome: Clinical, Histological, and Proteomic Findings of a 3-Year-Old Patient with Hereditary Gingival Fibromatosis. Biomedicines. 2019;7(3):48. Published 2019 Jun 29. doi:10.3390/biomedicines7030048
Staderini E, Guglielmi F, Cordaro M, Gallenzi P. Ossifying epulis in pseudohypo-parathyroidism: a case-based therapeutic approach. Eur J Paediatr Dent. 2018;19(3):218-220. doi:10.23804/ejpd.2018.19.03.9
Answer: we included this paragraph within the limitations of our study and included the recommended literature: “Moreover, we don't know if the self-consciousness of syndromic status may enhance the perception of any oral symptom as "abnormalities" or "invalidating factor". For instance, we don't know the clinical parameters (PPD, BoP, GI, PI) of the patients with "periodontitis,gingivitis"; therefore, we don't know if a normal person, with the same clinical parameters, would have the same OHIP-14 score as an EDS patient.” (line 211-215)
REFERENCES:
references should be formatted according to the journal guidelines.
Answer: we formatted the references according to the journal Guidelines.
Round 2
Reviewer 1 Report
In one spot, the authors refer to someone without out the condition of interest as a "normal person." That should be corrected.
I find their use of the term statistically noticeable confusing. Was it statistically significant or not. If no test of significance was performed, then they should say so.
Author Response
We would like to thank the Editor and the Reviewers for revising our manuscript [medicina-885699] entitled “Oral symptoms and oral health-related quality of life of people with Ehlers-Danlos syndromes (EDS): A questionnaire-based cross-sectional study“ and the constructive points discussed. The helpful comments and suggestions for improving the manuscript have been incorporated into the revised version and all changes were highlighted “Track Changes”. In this letter, we provide a point-by-point response to each addressed comment and hope the manuscript is now suitable for publication in Medicina.
Reviewer 1
Comments and Suggestions for Authors
In one spot, the authors refer to someone without out the condition of interest as a "normal person." That should be corrected.
Answer: we revised “normal person” to …”we don't know if a person who is not affected with EDS…” (line 216).
I find their use of the term statistically noticeable confusing. Was it statistically significant or not. If no test of significance was performed, then they should say so.
Answer: We used the term of “statistical noticeable” instead of “statistical significant” since – from a statistical point of few – the term “significant” is only meaningful in the context of a confirmatory study and should, strictly speaking, only be used there. Our study was purely explorative, and the results of the performed tests were thus interpreted accordingly.